Arthrological reconstructions of the pterosaur neck and their implications for the cervical position at rest

Buchmann Richard richardbuchmann@gmail.com 1 2
Rodrigues Taissa 1 2
1 Laboratório de Paleontologia, Departamento de Ciências Biológicas, Universidade Federal do Espírito Santo , Vitória , Espírito Santo , Brazil
2 Programa de Pós-graduação em Ciências Biológicas, Universidade Federal do Espírito Santo , Vitória , Espírito Santo , Brazil
Wedel Mathew
Electronic publication date: 2024 Feb 21
Publication date: 2024
Volume: 12
Electronic Location ID: e16884
Received 2023 Oct 6; Accepted 2024 Jan 12
Copyright: ©2024 Buchmann and Rodrigues
Copyright year: 2024
Copyright holder: Buchmann and Rodrigues
License: This is an open access article distributed under the terms of the Creative Commons Attribution License, which permits unrestricted use, distribution, reproduction and adaptation in any medium and for any purpose provided that it is properly attributed. For attribution, the original author(s), title, publication source (PeerJ) and either DOI or URL of the article must be cited.
License URL: https://creativecommons.org/licenses/by/4.0/

Keywords: Cervical biomechanics, Cervical vertebrae, Cervical ligaments, Pterosauria, Archosauria

Funding: Coordenação de Aperfeiçoamento de Pessoal de Nível Superior–Brazil (CAPES)–Finance Code 001 Conselho Nacional de Desenvolvimento Científico e Tecnológico (CNPq) #421412/2018-6 This study was funded by Coordenação de Aperfeiçoamento de Pessoal de Nível Superior–Brazil (CAPES)–Finance Code 001 (scholarship to Richard Buchmann) and by Conselho Nacional de Desenvolvimento Científico e Tecnológico (CNPq) (project grant #421412/2018-6 to Taissa Rodrigues). The funders had no role in study design, data collection and analysis, decision to publish, or preparation of the manuscript.

==============================
The lack of any pterosaur living descendants creates gaps in the knowledge of the biology of this group, including its cervical biomechanics, which makes it difficult to understand their posture and life habits. To mitigate part of this issue, we reconstructed the cervical osteology and arthrology of three pterosaurs, allowing us to make inferences about the position of the neck of these animals at rest. We used scans of three-dimensionally preserved cervical series of Anhanguera piscator, Azhdarcho lancicollis and Rhamphorhynchus muensteri for the reconstructions, thus representing different lineages. For the recognition of ligaments, joint cartilages, and levels of overlapping of the zygapophyses, we applied the Extant Phylogenetic Bracket method, based on various extant birds and on Caiman latirostris. We inferred that pterosaur intervertebral joints were probably covered by a thin layer of synovial cartilage whose thickness varied along the neck, being thicker in the posterior region. Ignoring this cartilage can affect reconstructions. According to the vertebral angulation, their neck was slightly sinuous when in rest position. Our analyses also indicate that pterosaurs had segmented and supra-segmented articular cervical ligaments, which could confer stabilization, execute passive forces on the neck and store elastic energy.

Introduction

Pterosaurs and birds are the only archosaur lineages of active fliers and have several converging characteristics, such as the presence of pneumatic bones, a well-developed sternum, and the fusion of some bones, which allows extrapolating functions from avian structures to those of pterosaurs (Bennett, 2001; Kellner & Campos, 2002; O’Connor, 2006; Butler, Barrett & Gower, 2012). These hypotheses can be tested through biomechanical studies (Rayfield, 2007; Lautenschlager, 2017). An understudied topic in pterosaur biology is the functional anatomy of the neck, despite the importance of cervical position in their biology and behavior (Zweers, Bout & Heidweiller, 1994; Witton & Naish, 2008; Averianov, 2013; Naish & Witton, 2017). With the adaptation of the forelimbs in the form of wings, the neck of birds and pterosaurs assumes the function of providing a movable skull, which allows the head to access resources in different locations (Marek et al., 2021; Marek, 2023).

In birds, the neck is divided into three functional segments, in which the first and second more cranial ones present greater freedom of movement in all axes (Boas, 1929; Zusi, 1962; Marek, 2023). The cervical vertebrae of pterosaurs also vary in their shape along the neck, which allows recognition of three anatomical segments (Bennett, 2001; Vila Nova et al., 2015). The atlas and axis (specialized elements for articulation with the skull) represent the more cranial one, the mid-cervical vertebrae (i.e., from the third to the seventh cervical vertebrae) constitute the second and occupy the middle of the neck, and two posterior cervical vertebrae at the caudal end are the third segment (Bennett, 2001; Vila Nova et al., 2015). Cervical anatomical variation is also easily perceived by comparing different species of pterosaurs, which can present characteristics ranging from short to extremely long mid-cervical vertebrae, from small to tall neural spines throughout the neck, presence to absence of cervical ribs, or unfused or fused atlas and axis (Wellnhofer, 1975; Wellnhofer, 1991; Kellner & Tomida, 2000; Bennett, 2001; Kellner, 2003; Averianov, 2010; Andres & Langston Jr, 2021). Such interspecific anatomical variations certainly have an impact on posture and biomechanics of the neck, which are still little investigated (Witton & Naish, 2008; Naish & Witton, 2017).

The cervical series supports a long skull especially in pterodactyloid pterosaurs, and some of these species also had large crests that represent additional weight to the head of the animal (Campos & Kellner, 1997; Kellner & Tomida, 2000; Kellner & Campos, 2002; Witton & Naish, 2008; Pinheiro et al., 2011; Bantim et al., 2014; Naish & Witton, 2017; Andres & Langston Jr, 2021 Beccari et al., 2021). Cervical posture is certainly influenced by the cranial anatomy of each species, considering that the neck receives stress resulting from the support and movement of frequently long and ornamented skulls (Campos & Kellner, 1997; Kellner & Campos, 2002; Witton & Naish, 2008; Pinheiro et al., 2011; Bantim et al., 2014; Naish & Witton, 2017; Andres & Langston Jr, 2021; Beccari et al., 2021). The posture of the neck in pterosaurs could still vary in relation to terrestrial or aerial locomotion, which could suffer from different mechanical impacts depending on the type of movement (Dzemski & Christian, 2007; Witton & Naish, 2013).

Furthermore, little is known about pterosaur neck arthrology, even though it influences the cervical angulation and freedom of movement of the neck in archosaurs (Tsuihiji, 2004; Tambussi et al., 2012; Taylor & Wedel, 2013; Taylor, 2014). Thus, the present study aims to reconstruct the cervical arthrology and the position of the pterosaur neck at rest, which could support more assertive proposals regarding cervical movements during locomotion and foraging. The position at rest considers a partial overlap of the zygapophyses, thus differing from the osteological neutral pose (ONP), which establishes total overlap of the zygapophyses (Stevens & Parrish, 1999). Partial overlapping with the neck at rest determine differences in terms of a more or less extended natural neck posture (Taylor, Wedel & Naish, 2009).

Materials & Methods

We analyzed the cervical vertebral column of the holotype of Anhanguera piscator (NSM-PV 19892; first-hand) (Kellner & Tomida, 2000), which comes from the Lower Cretaceous Romualdo Formation of the Santana Group (Araripe Basin, Brazil) and is housed in the collection of the National Museum of Nature and Science, in Tsukuba, Japan; of specimens attributed to Azhdarcho lancicollis (ZIN PH, several specimens; and CCMGE 1/11915 and 7/11915) (Averianov, 2010), from the Upper Cretaceous Bissekty Formation at Dzharakuduk, Uzbekistan and housed in the Paleoherpetological collection of the Zoological Institute of the Russian Academy of Sciences (ZIN PH) and Chernyshev’s Central Museum of Geological Exploration (CCMGE), both in Saint Petersburg, Russia; and a referred specimen of Rhamphorhynchus muensteri (MGUH 1891.738; first-hand) (Bonde & Christiansen, 2003) from the Upper Jurassic Solnhofen limestones of southern Germany and housed in the Geological Museum/Natural History Museum of Denmark, Copenhagen, Denmark. We chose these specimens due to the excellent three-dimensional preservation of almost all their cervical vertebrae.

Computed tomography (CT) scans of the Anhanguera piscator holotype were obtained by Takanobu Tsuihiji, Makoto Manabe and Chisako Sakata (National Museum of Nature and Science, Japan), at 300 to 310 kV and 200 µA, with different voxel sizes for different scans (Table 1). The complete plate containing R. muensteri (MGUH 1891.738) was scanned by Niels Bonde and Maria Eduarda Castro Leal (Zoological Museum, Denmark), at 120 kV and 280 µA, and the voxel sizes were 0.2 mm. The cervical vertebrae and notarium of Azhdarcho lancicollis (ZIN PH; several specimens and CCMGE 1/11915 and 7/11915) were digitalized by noncontact laser 3D-scanning and given to this research by Alexander Averianov (Zoological Institute of the Russian Academy of Sciences, Russia).

Table 1 Voxel sizes for different scans of the Anhanguera piscator holotype.

Element	Voxel size (mm)	
Skull	0.198	
Atlas + axis and cervical III	0.065	
Cervical IV	0.075	
Cervical V	0.175	
Cervical VII	0.166	
Cervical VIII	0.166	
Cervical IX	0.166	
Dorsal I	0.166	

We made osteological reconstructions of the damaged parts of the vertebrae and included soft tissues using the Blender 3D software, version 2.91 (Blender Development Team, 2019). The vertebrae of R. muensteri are attached to a slab that displays the cervical series in ventral view on the main surface of the slab and the neural spines on the opposite surface. They were digitally isolated from the matrix and reconstructed based on previously published photographs and drawings of other specimens of this species (Wellnhofer, 1975). The cervicals of Anhanguera piscator were already prepared from the rock before scanning, but as its sixth vertebra was not preserved, we reconstructed it based on the sixth vertebra of AMNH 22555, belonging to Anhanguera sp. (Wellnhofer, 1991; Pinheiro & Rodrigues, 2017), due to the similarity of their other preserved elements. We had to make more inferences for the osteological reconstruction of the cervicals of Azhdarcho lancicollis, due to the lack of completely preserved specimens. The well-defined regionalization of the cervical vertebrae of Anhanguera piscator, R. muensteri and other pterosaurs, including other azhdarchids (Bennett, 2001; Pereda-Suberbiola et al., 2003; Kellner, 2010), indicates that Azhdarcho lancicollis probably also had mid-cervical and posterior cervical vertebrae with distinct morphologies. The differences are more noticeable in their lengths, which are very long in mid-cervicals. All three-dimensional models attributed to the cervical vertebrae used in this work are available on Morphosource (Boyer et al., 2017). The main plate containing specimen MGUH 1891 (Rhamphorhynchus muensteri; Media ID 000588063) is available on Morphosource under the DOI 10.17602/M2/M588063. 3D models of the vertebrae of specimen NSM-PV 19892 (Anhanguera piscator) are available on Morphosource under DOI 10.17602/M2/M589278 (Media ID 000589278), DOI 10.17602/M2/M589282 (Media ID 000589282), DOI 10.17602/M2/M589286 (Media ID 000589286), DOI 10.17602/M2/M589290 (Media ID 000589290), DOI 10.17602/M2/M589294 (Media ID 000589294), and DOI 10.17602/M2/M589298 (Media ID 000589298). 3D models of the vertebrae of Azhdarcho lancicollis are available on Morphosource under ARK identifiers DOI 10.17602/M2/M568148 (specimen ZIN PH 105/44; Media ID 000568148), DOI 10.17602/M2/M568153 (ZIN PH 131/44; Media ID 000568153), DOI 10.17602/M2/M568158 (ZIN PH 144/44; Media ID 000568158) , DOI 10.17602/M2/M568163 (CCMGE 1/11915 and ZIN PH 139/44; Media ID 000568163), DOI 10.17602/M2/M568168 (ZIN PH 147 /44; Media ID 000568168), DOI 10.17602/M2/M568173 (ZIN PH 138/44; Media ID 000568173), DOI 10.17602/M2/M568178 (ZIN PH 137/44; Media ID 000568178), DOI 10.17602/M2/M568183 (ZIN PH 122/44; Media ID 000568183) and DOI 10.17602/M2/M568188 (CCGME 7/11915; Media ID 000568188).

We reconstructed cervical arthrology based on the Extant Phylogenetic Bracket (EPB), which establishes criteria for soft tissue inferences from the anatomy of extant outgroups that are phylogenetically close to the group being studied (Witmer, 1995). Inference level I occurs when a given soft tissue is associated with an osteological correlate present in both outgroups and such correlate is also present in pterosaurs. Inference level I’ is established if soft tissue is present in both outgroups but without any evidence of an osteological correlate in pterosaurs. Inference level II is obtained when the soft tissue is associated with a structure in only one of the outgroups and the osteological correlate is present in pterosaurs. Inference level II’ is configured when a certain soft tissue is present in only one of the external groups and we have not identified the osteological correlate in pterosaurs. Inference level III is considered when the osteological structure that suggests the presence of soft tissue in pterosaurs was not seen in any of the outgroups. Finally, inference level III’ does not involve any osteological evidence (Witmer, 1995). As outgroups, we obtained data first-hand by dissecting a broad-snouted caiman (Caiman latirostris) and seventeen extant birds, of which eleven were Phaethoquornithes (Ardea alba, Calonectris diomedea, Fregata magnificens, Ixobrychus exilis, Nannopterum brasiliensis, Phaeton aethereus, Procellaria aequinoctialis, Puffinus puffinus, Sula leucogaster, Thalassarche chlororhynchos and Thalassarche melanophris), four Charadriiformes (Haematopus palliatus, Larus dominicanus, Sterna hirundo and Thalasseus acuflavidus) and two were Telluraves (Coragyps atratus and Cariama cristata) (Prum et al., 2015; Reddy et al., 2017; Kuhl et al., 2021; Braun & Kimball, 2021). Phaethoquornithes and Charadriiformes have foraging habits associated with coastal or freshwater environments, which is thought to be similar to several pterosaurs (Sick, 1997; Bennett, 2001; Schreiber & Burger, 2001; Kellner & Campos, 2002). The analyzed Telluraves have more terrestrial habits, which have also been inferred for some azhdarchids (Witton & Naish, 2008; Naish & Witton, 2017). The dissected avian specimens died or were euthanized after veterinary monitoring at the Institute for the Research and Rehabilitation of Marine Animals (IPRAM), in Cariacica/ES, Brazil, which is responsible for the rehabilitation of marine animals under the authorizations SISFAUNA IEMA 001/2014, process 68077610; IEMA 001/2014, process 67277780; and SISBIO 34510 and 26896. The caiman was euthanized at the Caiman Project headquarters (under the coordination of the Marcos Daniel Institute, a non-profit organization), in Vitória/ES, Brazil, under the authorization SISBIO 92997549, and dissected at the Veterinary Hospital of the Universidade Federal do Espírito Santo, in Alegre/ES, Brazil.

We measured the length, height and width of the vertebrae of Anhanguera piscator and Rhamphorhynchus muensteri first-hand with a caliper. The same measurements for Azhdarcho lancicollis were taken from Averianov (2010) or measured in the MeshLab software, version 2021.10 (Cignoni et al., 2008), on the 3D scan images. The angles between the vertebrae were also measured with Blender 3D software, version 2.91 (Blender Development Team, 2019), and were determined by the position of the vertebral axis in relation to the axis of its preceding vertebra. The distance between the condyle and the cotyle of adjacent vertebrae was measured in the central and peripheral regions of the joint. The area of the zygapophyses was measured only in Anhanguera piscator and Azhdarcho lancicollis, due to the vertebrae of R. muensteri being preserved articulated. Both these measurements were obtained using the ImageJ software (Schneider, Rasband & Eliceiri, 2012). We used the nomenclature by Baumel & Witmer (1993) for the osteological and arthrological descriptions.

The partial articulation between the zygapophyses indicates the presence of gaps between the condyle and the cotyle of the adjacent cervical vertebra, which would be filled by synovial cartilage, annular ligaments, and fluid (Baumel & Raikow, 1993; Taylor & Wedel, 2013). The increase in cervical length due to the presence of vertebral cartilage was calculated by subtracting the total length of the articulated vertebrae with partial overlapping zygapophyses from the sum of the lengths of all cervical vertebrae and per neck segment (Taylor & Wedel, 2013).

Although the atlas and axis of R. muensteri are not fused as in the other analyzed pterosaurs, we disregarded the span length in this joint because, due to the absence of zygapophyses and to the restrict movement between these vertebrae, the addition of cartilage would be irrelevant for the analysis of the position at rest and to the degree of freedom to which the neck would be subject.

Results

Neck posture and ligaments of extant archosaurs

The facies articularis cranialis and caudalis of all analyzed birds is heterocoelic, and procoelic in Caiman latirostris. In all specimens analyzed, the intervertebral space has synovial cartilage between the cervical vertebrae, which is associated with well-defined rough areas at the margins of the facies articularis (Figs. 1A–1D). The intervertebral meniscus is bathed in the synovial fluid, providing flexibility to the joint. The synovial cartilage is limited by an articular capsule, allowing flexibility in the sagittal and frontal planes. The dorsal portion of the facies articularis caudalis of all specimens shows a small fovea (Figs. 1C, 1D).

Figure 1 Photographs and interpretative drawings of the cervical vertebrae of extant birds.

Eighth vertebra belonging to Procellaria aequinoctialis in dorsal view (A, B) and caudolateral view (C, D). Cervical series from the tenth to thirteenth vertebra belonging to Procellaria aequinoctialis (E, F). Dashed line indicates reconstructed regions. Dotted regions indicate rougher surfaces. Scale bars: 10 mm.

The length of the intervertebral space including the articular tissues is often less than 0.5 mm in all specimens, representing between 2 and 6% of the total length of two articulated vertebrae. The exception is closer to the base of the neck (i.e., posteriorly), in which the length of the intervertebral joint is greater than 0.5 mm and represents 8 to 10% of the length of two articulated vertebrae. However, when we consider the complete cervical series, the cartilage represents between 3 and 6% of the total length of the neck of the analyzed birds and caiman.

Synovial cartilage is also present in the zygapophyseal articulations, where we also observe a rough surface at the edges of the facets. In this joint, the synovial fluid allows the thin menisci to slide between the facets of the prezygapophyses and postzygapophyses, limited by extremely malleable articular capsules. There is a partial overlap of the zygapophyses at rest position, from approximately 50% or more (Figs. 1E, 1F). The greatest overlaps are on zygapophyseal joints close to the base of the neck (Figs. 1E, 1F). Manipulating the specimens to a complete cervical dorsoventral flexion result in very low or absent overlap of the zygapophyses in the cranial half of the neck, and approximately 50% near the base.

We observe cervical ligaments of extant archosaurs that are associated with rough surfaces, cortical grooves and/or protuberances of the vertebrae. The ligamentum collaterale attaches to a rough-surfaced tubercle of the laterocaudal portion of the centrum and inserts on a lateroventral tubercle of the ansa costotransversaria of the succeeding vertebra in birds (Figs. 2A, 2B). In C. latirostris, the ligamentum interarticulare also attaches to the laterocaudal portion of the centrum and inserts on the smooth surface of the lateral tubercles of the neural arch, but it is arranged more vertically due to the vertebral anatomy of the caiman (Fig. 2E). Both ligaments border the intervertebral joints from the axis to the end of the cervical series in all specimens analyzed. In birds, the ligamentum ventrolaterale attaches to the rough surfaces of the carotid processes of the vertebrae of the second cervical segment, and the ligamentum intercristale ventrale, to the rough area of the hypapophyses of the vertebrae of the third neck segment. In C. latirostris, the ligamentum intercostale attaches to ventral rough surfaces at the ends of each cervical rib along the neck (Fig. 2E).

Figure 2 Photographs and interpretative drawings of cervical series of extant birds and Caiman latirostris.

Cervical series from the eighth to the thirteenth vertebra belonging to Haematopus palliatus in dorsal view (A, B); cervical series from the ninth to the fourteenth vertebra belonging to Procellaria aequinoctialis in left lateral view (C, D); interpretative drawing of the cervical series of Caiman latirostris showing the position of the cervical ligaments (E). Dotted regions indicate rougher surfaces. Scale bars: 10 mm.

In the neural arch, we observe roughness at the base of the neural spine of birds and C. latirostris, which marks the attachment site of the ligamentum elasticum interlaminare (Figs. 2C, 2D). This ligament is absent between the vertebrae of the second segment of birds that have an extremely reduced neural spine (Figs. 2C, 2D). In C. latirostris, we also observed laterocranial and laterocaudal grooves and roughness at the top of the neural spine, which related to the attachment sites of the ligamentum supraspinale, a segmented ligament that connects adjacent vertebrae (Fig. 2E).

Reconstruction of the pterosaur neck

All analyzed vertebrae of Azhdarcho lancicollis have preserved prominent hypapophysis and postexapophyses, except the seventh, which does not have a well-preserved centrum. The neural spines of the mid-cervical vertebrae of this azhdarchid are known by fragmented parts, mainly from the third and seventh vertebrae (ZIN PH 131/44; ZIN PH 138/44). We observed that they are tallest at the cranial and caudal ends on the fourth, fifth, and sixth vertebrae (ZIN PH 144/44; CCMGE 1/11915, ZIN PH 139/44; ZIN PH 147/44), with the middle part of these vertebrae having a low spine and thus a cylindrical appearance. The seventh vertebra (ZIN PH 138/44) does not have a preserved neural spine but the latero-dorsal laminae of the neural arch are directed towards the mid-dorsal surface along the entire length, giving the neural arch a triangular cross-section, different from the more cranial vertebrae. This architecture suggests a neural spine of uniform height along the seventh vertebra (Fig. 3), but the dimensions of the neural arch are not consistent with the presence of a tall neural spine, and it apparently does not exceed the maximum height of the neural spine of the sixth vertebra (ZIN PH 147/44).

Figure 3 CT scans and interpretative drawings of the cervical vertebrae of Azhdarcho lancicollis.

Seventh vertebra (A, B) in left lateral view and the eighth vertebra (C, D) in left dorsolateral view (ZIN PH 138/44 and ZIN PH 137/44, respectively) of Azhdarcho lancicollis. Dashed line indicates reconstructed regions.

The posterior cervical vertebrae of Azhdarcho lancicollis are more poorly preserved. The eighth vertebra (ZIN PH 137/44) has a short centrum and a neural arch whose dorsal region is completely lost (Fig. 3C). Besides a fragmented and short centrum, the ninth vertebra (ZIN PH 122/44) still has part of its neural arch preserved, suggesting a tall neural spine. We thus interpret that the neural spines of both posterior cervicals of this pterosaur are taller than those of the mid-cervicals (Fig. 3D). This morphology is similar to that of the dorsal vertebrae, as also observed in the posterior cervicals of Anhanguera piscator and Rhamphorhynchus muensteri (Figs. 4E, 4F).

Figure 4 Photographs and interpretative drawings of the pterosaur cervical vertebrae.

Fourth vertebra (A, B) in caudal view, fifth vertebra (C, D) in cranial view and eighth vertebra (E, F) in caudal view of Anhanguera piscator (NSM-PV 19892) and the articulated seventh and eighth vertebrae (G, H) in ventrolateral view of Rhamphorhynchus muensteri (MGUH 1891. 738). Dashed line indicates reconstructed regions. Dotted regions indicate rougher surfaces. Scale bars: 10 mm.

In the analyzed pterosaurs, the mid-cervical vertebrae are either long and tall, as in Anhanguera piscator, or extremely long, as in Azhdarcho lancicollis (Table S1), while the posterior cervicals of all pterosaurs present a short centrum, tall neural spine and broad transverse processes (Table S1).

The cervical vertebrae of the three analyzed pterosaurs are procoelic and exhibit an extremely convex condyle (Fig. 4). The presence of synovial cartilage in the joints is suggested by the rough areas mainly on the margins of the articular surfaces of the zygapophyses, cotyle and condyle (Fig. 4). There is also a small dorsomedial fovea on the condyle, which is more robust in the posterior cervical vertebrae (Fig. 4). The presence of these osteological correlates in the birds and caiman analyzed allows a level I EPB inference for the presence of synovial cartilage in pterosaur intervertebral joints (Table 2).

Table 2 Soft tissues inferred for the cervical spine of pterosaurs.

Relationship of inferred soft tissues to the sites at which they attach in the cervical vertebrae of extant archosaurs, the osteological correlates in pterosaur cervical vertebrae, and the level of inference by EPB.

Soft tissue	Attachment site in crocodylians	Attachment site in birds	Osteological correlates in pterosaur	Inference level by the EPB	
Synovial cartilage in the joints	Fovea in the condyle and roughness on the edges of the articular surfaces	Fovea in the condyle and roughness on the edges of the articular surfaces	Fovea in the condyle and roughness on the edges of the articular surfaces	I	
Ligamentum collaterale	The laterocaudal portion of the centrum and the smooth surface of the lateral tubercles of the neural arch	Rough-surfaced tubercle of the laterocaudal portion of the centrum and lateroventral tubercle of the ansa costotransversaria	The rough and laterally developed postexapophyses and tubercles on the ansa costotransversaria	I	
Ligamentum elasticum interlaminare	Roughness at the cranial and caudal bases of the neural spine	Roughness at the cranial and caudal bases of the neural spine	Roughness at the cranial and caudal bases of neural spines	I	
Ligamentum supraspinale	Laterocranial and laterocaudal grooves and roughness at the top of the neural spine	Absent	Lateral grooves and rough areas from the dorsocranial to the dorsocaudal surface of the cervical neural spines	II	
Ligamentum nuchae	Absent	Absent	Roughness at the tops of the neural spines	III	

Although the atlas-axis and the third vertebra are still articulated in the Anhanguera piscator holotype, we cannot be sure that the angle and distance between them, as preserved, are similar to the position of the neck at rest. We infer here a level of overlap of the zygapophyses of approximately 50% for the joints between the axis and the mid-cervical vertebrae and approximately 75% for those in the posterior cervicals in all pterosaurs analyzed. This partial overlapping of the zygapophyses is similar to the position of the neck at rest in extant archosaurs. This overlapping and the anatomy of the pterosaur vertebrae allow inferences regarding neck sinuosity. The short postzygapophyses and the less convex condyle of the third vertebra compared to the other mid-cervicals indicate that it articulated more ventrally in relation to the axis, as is also the orientation of the fourth cervical in relation to the third vertebra (Table 3). This arrangement suggests a downward curvature in relation to the atlas and axis (Fig. 5). The articulated series of the mid-cervical vertebrae (i.e., post-fourth vertebra) is arched dorsally, conferring sinuosity to the necks (Fig. 4). In Anhanguera piscator and R. muensteri, the second largest angle between the neck joints is between the sixth and seventh cervicals (Table 3). In Azhdarcho lancicollis, the largest angle between the mid-cervical vertebrae is more cranial, between the fifth and sixth vertebrae. The cervical series is straighter in the posterior segment of the neck of all analyzed pterosaurs, based on the orientations of the joints between consecutive vertebrae (Table 3). The difference between the mid and posterior regions of the neck was also due to the longer prezygapophyses and the more convex condyle of the mid-cervicals than the posterior cervicals. The angle observed between the joints and the lengths of the vertebrae makes the neck of Azhdarcho lancicollis more vertical than that of the other pterosaurs (Fig. 5). Our three-dimensional models corresponding to the reconstructions of the cervical series at rest are available on Morphosource under the following DOI 10.17602/M2/M588059 (Rhamphorhynchus muensteri, based on the MGUH specimen 1891; Media ID 000588059), DOI 10.17602/M2/M589271 (Anhanguera piscator, based on specimen NSM-PV 19892; Media ID 000589271) and DOI 10.17602/M2/M59913 (Azhdarcho lancicollis, based on multiple specimens; Media ID 000599131).

Table 3 Angles of inclination and distance of the condyle and cotyle between a vertebra and its succeeding one on the neck of pterosaurs at rest.

(+) above and (-) below the axis of succeeding vertebra. Roman algorisms indicate each vertebra’s position.

	Angle	Central distance (mm)	Peripheral distance (mm)	
Anhanguera piscator				
ATAX–CV III	−22°	2.02	2.47	
CV III–CV IV	−6°	1.98	2.48	
CV IV–CV V	+7°	2.15	2.22	
CV V–CV VI	+6°	2.04	2.62	
CV VI–CV VII	+19°	3.33	3.94	
CV VII–CV VIII	+10°	3.67	4.62	
CV VIII–CV IX	−6°	4.49	4.76	
CV IX–DV I	−10°	5.24	5.64	
Azhdarcho lancicollis				
ATAX–CV III	−46°	0.79	1.67	
CV III–CV IV	−14°	0.95	1.28	
CV IV–CV V	+12°	0.98	1.31	
CV V–CV VI	+16°	1.04	1.25	
CV VI–CV VII	+5°	1.33	1.78	
CV VII–CV VIII	+7°	2.2	2.59	
CV VIII–CV IX	+20°	2.46	2.62	
CV IX–DV I	−19°	2.14	2.25	
Rhamphorhynchus muensteri				
ATAX–CV III	−17°	0.11	0.13	
CV III–CV IV	−2°	0.08	0.11	
CV IV–CV V	+2°	0.10	0.11	
CV V–CV VI	+6°	0.29	0.32	
CV VI–CV VII	+12°	0.32	0.36	
CV VII–CV VIII	−4°	0.38	0.48	
CV VIII–CV IX	+3°	0.42	0.44	
CV IX–DV I	+6°	0.43	0.49	
Notes.

AbbreviationsATAX atlas and axis

CV cervical vertebra

DV dorsal vertebra

Figure 5 Reconstruction of the position at rest of the cervical vertebral column of the pterosaurs.

Cervical series of Anhanguera piscator (A), Azhdarcho lancicollis (B) and Rhamphorhynchus muensteri (C) in left lateral view. Inset on the upper right corner shows a reconstructed Anhanguera piscator skull and neck. The intervertebral cartilage and cartilage of the zygapophyseal joints are shown in light brown. Scale bars: 10 mm.

In all analyzed pterosaurs, the prezygapophyses of the cervical vertebrae have their facets facing dorsally and slightly cranially, and oriented towards the medial side, while the opposite is observed in the postzygapophyses, whose facets are pointed ventrally, slightly caudally and turned laterally to the vertebrae. This allows the zygapophyses, partially superimposed in rest position, to slide their cartilage on top of each other in all directions, though limited by an articular capsule, with friction reduced by the synovial fluid (Fig. 6). The only exception are the postzygapophyses of the axis, which face entirely ventrally, especially in Azhdarcho lancicollis, demonstrating that the movement in the articulation with the prezygapophyses of the third vertebra is possibly more sagittal, while the other cervicals are likely to slip slightly diagonally. Besides the orientation, the facets of the zygapophyses are larger in the middle portion of the neck. Comparing the different species, the zygapophyses of Azhdarcho lancicollis are slightly longer, increasing the craniocaudal distance they can slide over each other and likely impacting the neck’s range of motion. As a result of the articulation of the cervical series with the partial overlap of the zygapophyses, the gaps between condyles and cotyles indicate that the intervertebral cartilage corresponds to 2.1%–6.5% of the total neck length (Table 4).

Figure 6 Reconstruction of the articular capsule and cervical ligaments of the neck of pterosaurs at rest position.

Cervical series of Anhanguera piscator (A), Azhdarcho lancicollis (B) and Rhamphorhynchus muensteri (C) in left lateral view.

Table 4 Proportion of total cartilage in the neck length.

	CV (mm)	TOTAL (mm)	CART (%)	
Anhanguera piscator	353.4	378.3	6.5	
Azhdarcho lancicollis	539.6	551.4	2.1	
Rhamphorhynchus muensteri	61.9	64.1	4.0	
Notes.

Abbreviations CV sum of the centrum lengths of the cervical vertebrae

TOTAL total neck length

CART percentage of synovial cartilage on the neck centra

The shorter pre- and postzygapophyses in the posterior cervicals indicate that the cartilage present in the intervertebral joint in this region are thicker than in the more cranial portions of the neck (Table 3; Fig. 7). In the latter, the prezygapophyses elongate cranially and exceed the length of the condyle. Due to the very convex condyle, the synovial cartilage in the peripheral parts of the joints is probably thicker than in the center (Table 3).

Figure 7 Reconstruction showing the probable variation in synovial cartilage thickness along the neck of pterosaurs.

Cervical series of Anhanguera piscator (A) and Rhamphorhynchus muensteri (B) in ventral view, with the anterior and posterior portions of the neck in detail (middle and right-side images, respectively).

Although the ligaments of the pterosaur neck are difficult to understand, they probably had the ligamentum collaterale and segmented ligaments in the neural arch. The laterally developed postexapophyses with rough surfaces along the entire neck in all analyzed pterosaurs suggest the attachment site of the ligamentum collaterale (Fig. 4). The tubercles lateral to the ansa costotransversaria indicate its insertion, which possibly behaved as an extension of the articular capsule and contributed to joint stiffness throughout the neck (Figs. 6 and 7). Based on the identified osteological correlates in extant archosaurs, the ligamentum collaterale of birds is probably homologous to the ligamentum interarticulare of Caiman latirostris. Therefore, we consider the presence of this ligament in the pterosaur neck a level I EPB inference (Table 2).

The elevated head in relation to the neck, as observed in our reconstructions of the position at rest (Fig. 5), hints the presence of segmented stabilizing elastic ligaments dorsally to the cervical vertebrae (Fig. 6). An accentuated rough area at the base of the neural spines along the necks of the analyzed pterosaurs implies the presence of the ligamentum elasticum interlaminare. The presence of this correlate in the cervical vertebrae of the analyzed birds and caiman indicates that this is a level I EPB inference (Table 2). The absence of this correlate in the extremely reduced neural spines of the mid-cervical vertebrae of Azhdarcho lancicollis suggest a peculiar absence of this ligament in the middle portion of the neck (Fig. 6).

We observed lateral grooves and rough areas from the dorsocranial to the dorsocaudal surface of the cervical neural spines of the analyzed pterosaurs. Osteological correlates support the inference of a segmented ligament, such as the ligamentum supraspinale seen in Caiman latirostris. However, the presence of a medial groove medially and extensive rough areas at the top of the neural spine may also indicate the ligamentum nuchae, which was not observed in the extant archosaurs analyzed. The presence of the ligamentum nuchae represents a level III EPB inference, while the presence of the ligamentum supraspinale represents a level II EPB inference according to the correlates seen in extant archosaurs (Table 2). The presence of grooves and roughness at the top of the neural spine of the most cranial dorsal vertebrae of Anhanguera piscator and R. muensteri suggests that both ligaments were not limited to the neck. Alternatively, the presence of muscle scars in the medial portion at the top of the neural spine may also contribute to excessive roughness. The constant increase in the width of the top of the neural spine from the mid-cervicals to the posterior cervical vertebrae supports the hypothesis that both ligaments are thinnest in the mid-portion of the neck and get gradually thicker towards the base of the neck. In addition, a ligamentum nuchae could vary in height and present itself as an additional surface of muscle attachment when associated with vertebrae with low neural spines (Fig. 6).

Discussion

Due to differences in vertebral morphologies from one neck region to the other, transitional vertebral morphologies are often present between cervical regions and between the cervical and dorsal vertebrae of archosaurs (Bennett, 2001; Tambussi et al., 2012; Iijima & Kubo, 2019). For instance, the first set of vertebrae posterior to the neck of birds are commonly named cervico-dorsales (Baumel & Witmer, 1993).

In the analyzed pterosaurs, the morphology of the mid-cervical vertebrae ranged from long or extremely long with low or developed neural spines, consistent with what is widely observed in Pterodactyloidea (Howse, 1986; Wellnhofer, 1991; Kellner & Tomida, 2000; Bennett, 2001; Kellner, 2003; Pereda-Suberbiola et al., 2003; Ösi, Weishampel & Jianu, 2005; Averianov, 2010; Henderson & Peterson, 2006; Beccari et al., 2021; Andres & Langston Jr, 2021). In Rhamphorhynchidae, neural spines are proportionately tall in all mid-cervical vertebrae, which are slightly longer than the posterior cervical vertebrae (Wellnhofer, 1975; Andres, Clark & Xing, 2010). The variation in the height of the neural spines in the mid-cervicals seen among pterosaurs imply differences in the locations of soft tissue attachments and in the mobility of the neck. The tall neural spines of the mid-cervical vertebrae of Anhanguera piscator and Rhamphorhynchus muensteri, together with the long prezygapophyses, indicate a certain level of restriction to dorsal extension in this portion of the neck, suggesting that this region might not extend much beyond the position at rest (Witton & Naish, 2008; Molnar et al., 2015). However, the tall vertebrae presented throughout the neck are adequate to conflict with the large bending moment probably generated by the long skull of these pterosaurs, which would stimulate the transmission of compression stresses in the centrum and tension in the neural arch of the cervical vertebrae during ventral flexion (Christian & Preuschoft, 1996; Tambussi et al., 2012).

The long centrum of mid-cervicals is the main mechanism of neck elongation in pterosaurs, as also seen in extant birds and in non-avian dinosaurs, although they also have more cervical vertebrae (Taylor & Wedel, 2013; Marek et al., 2021). The lengthening of the mid-cervical vertebrae indicates an increase in the articular lever arms, which probably contributed to the optimization of dorsoventral flexion throughout the neck (Christian & Preuschoft, 1996). However, this strategy of increasing the lever arms in the neck of pterosaurs does not seem to guarantee more flexibility than in the neck of birds, in which the increase in intervertebral joints guarantees more centers of rotation and the arrangement in functional segments guarantees long lever arms (Boas, 1929; Zusi, 1962; Dzemski & Christian, 2007; Cobley, Rayfield & Barrett, 2013). On the other hand, the intervertebral procoelic joints possibly contributed little to the joint flexibility (Fronimos, Wilson & Baumiller, 2016; Fronimos & Wilson, 2017). The procoelic vertebrae provided stability to the center of rotation during movement, due to the concave surface firming the convex surface of the joint in the opposite direction to the muscular stimulus (Dzemski & Christian, 2007; Molnar et al., 2015; Fronimos, Wilson & Baumiller, 2016). The extensive convexity of the condyle of the mid-cervical vertebrae could be related to the high shear stresses that this region of the neck could be subjected to Fronimos & Wilson (2017).

In all pterosaurs analyzed, the posterior cervical vertebrae have tall neural spines and lateral bone extensions with diapophyses, related to transverse processes, which may be homologous to those observed in the dorsals of other archosaurs (Wellnhofer, 1975; Bennett, 2001; Souza, 2018). The presence of a tall neural spine in posterior cervicals in Azhdarcho lancicollis agrees with other azhdarchid and azhdarchoid pterosaurs (Pereda-Suberbiola et al., 2003; Aires et al., 2014, Vila Nova et al., 2015; Buchmann et al., 2018; Andres & Langston Jr, 2021). The craniocaudally shorter centrum in posterior cervicals than in other cervicals is also observed in extant birds and crocodylians (Frey, 1988; Iijima & Kubo, 2019; Terray et al., 2020). As in the mid-cervicals, longer or taller processes in the posterior cervicals may limit mobility in this region of the neck, even though those structures are compatible with osteological correlates for the origin of cervical muscles (Boas, 1929; Witton & Naish, 2008; Tambussi et al., 2012; Iijima & Kubo, 2019). Furthermore, the high neural spines on posterior cervical vertebrae are possibly due to the mechanical need to combat the bending moment produced during dorsoventral flexion, which would probably impact this region more as it is close to the base of a long neck (Christian & Preuschoft, 1996; Tambussi et al., 2012).

The pterosaur anatomical vertebral segments we recognize here agree with Bennett (2001) and are widely recognized in extant birds and crocodylians (Terray et al., 2020; Mook, 1921; Iijima & Kubo, 2019). Defining these regions is important to recognize possible attachment sites of muscle and/or ligament tissue in extinct archosaurs (Wedel & Sanders, 2002; Tsuihiji, 2004; Tsuihiji, 2005; Tsuihiji, 2007; Snively & Russell, 2007). Although the archosaur cervical anatomical segments do not entirely agree with functional segments (Kambic, Biewener & Pierce, 2017), they are related to the muscles’ anatomy and morphology, which in turn are associated to varied locomotor and foraging habits in extant species (Molnar et al., 2015; Marek et al., 2021). Therefore, the morphological variation between cervical vertebrae in the analyzed pterosaurs, especially in the azhdarchid, reflect differences in that animal’s life habits in relation to the Anhanguera and Rhamphorhynchus (Averianov, 2013; Witton & Naish, 2008; Witton & Naish, 2013; Naish & Witton, 2017).

The procoelic pterosaur cervicals differ from the heterocelic morphology of extant birds and resemble the morphology of extant crocodylians, in which they constitute joints that guarantee stiffness to the neck while maintaining a large range of motion (Baumel & Witmer, 1993; Molnar et al., 2015). The variation in convexity of the condyles and the length of the zygapophyses between the cervical segments probably cause different movements along the neck in range of motion analyses of extinct animals, since the bone intersection between these structures and adjacent vertebrae are the main factors that limit movement (Jones, Brocklehurstm & Pierce, 2021; Padian et al., 2021). However, disregarding the presence of soft tissue during manipulation of the vertebrae tends to reveal a range of motion that was not necessarily performed by the animal in life (Hutson & Hutson, 2012; Padian et al., 2021).

The convex condyle of the pterosaur cervical vertebrae makes a precise fit to the cotyle of its neighboring vertebra, especially in the mid-cervicals, as observed in the measurements of the intervertebral space (Table 2) and supports the hypothesis of a thin layer of cartilage covering this gap. Furthermore, the anatomical differences observed between the hypapophysis of pterosaurs and crocodylians and the presence of preexapophyses and posexapophyses, which originate directly from the ventral portion of the cotyle and condyle, respectively, suggest that pterosaurs did not have a fibrous ring in the joint, or that, if present, it would be thinner than in extant crocodylians (Salisbury & Frey, 2001; Fronimos & Wilson, 2017).

Considering cartilage and ligaments allows for more assertive inferences on range of motion throughout the neck, as demonstrated in extant birds (Boas, 1929; Zusi, 1962; Dzemski & Christian, 2007; Molnar et al., 2015). The compatibility of the presence of synovial cartilage with the dorsomedial fovea (fovea condyli) and of flattened ventral edges in relation to the central region of the condyle has previously been reported in extant crocodylians (Salisbury & Frey, 2001; Fronimos & Wilson, 2017). Roughness on extant archosaur vertebral joint surfaces support our inference for the presence of synovial cartilage in pterosaurs, although some roughness is also found in mammalian vertebrae, which have intervertebral discs (McGowan, 1986; Fronimos & Wilson, 2017).

The rough edges of the zygapophyses, cotyle and condyle in the analyzed pterosaur vertebrae also support inferences on the presence of a fibrous articular capsule connected to the synovial cartilage in these joints, also present in extant archosaurs (Schwarz, Frey & Meyer, 2007; Fronimos & Wilson, 2017). The presence of the joint capsule in extant birds fixes the joint and can allow both facets to slide until the zygapophyses are nearly disarticulated (Taylor, Wedel & Naish, 2009; Cobley, Rayfield & Barrett, 2013; Kambic, Biewener & Pierce, 2017). Establishing the level of overlap of the zygapophyses is also important before submitting a 3D model to range of motion analysis, so that the level of disarticulation is measured from the neck position at rest (Jones, Brocklehurstm & Pierce, 2021).

Our estimates of intervertebral cartilage thickness are based on the intervertebral space that defines the length of synovial cartilage in extant animals (Taylor & Wedel, 2013; Taylor, 2014; Vidal et al., 2020). This cartilage adds length to the neck, but that can vary according to the animal’s ontogenetic stage, as it may reduce during life (Vidal et al., 2020). The proportion of cartilage to the total neck length of the analyzed pterosaurs differs from that observed in extant mammals and crocodylians, which have more than 10% of the neck length composed of cartilage (Taylor & Wedel, 2013). This difference was expected due to the presence of intervertebral discs in the joints of extant mammals and some crocodylians, which are more robust than the likely thin synovial cartilages in pterosaurs (Taylor & Wedel, 2013; Vidal et al., 2020).

The proportion of cartilage we estimated for Azhdarcho and Rhamphorhynchus is similar to that observed in the Neognathae birds (Heidweiller, 1989; Buchmann, 2022). As in Azhdarcho lancicollis, extant birds with extremely long and dorsoventrally compact vertebrae have less intervertebral cartilage in the neck, as Ardea alba and Ixobrychus exilis (Buchmann, 2022). The relatively thin intervertebral articular capsules proposed for Azhdarcho lancicollis corresponds to the intervertebral soft tissue thickness previously proposed in a range of motion analysis of the vertebral series of another azhdarchid (Padian et al., 2021). The estimated 6.5% proportion of cartilage that makes up the neck length in Anhanguera piscator is similar to that observed in ostriches (6.3%), despite the presence of a fibrous ring in ostrich joints, which increase the thickness of intervertebral space (Cobley, Rayfield & Barrett, 2013).

Studies with Sauropodomorpha vertebrae also indicated large differences in this proportion, but they may also occur due to ontogenetic factors or due to the inaccuracy of the neck posture at rest (Taylor & Wedel, 2013; Taylor, 2014). Although the proportion of cartilage obtained for pterosaurs reveals a smaller amount compared to extant mammals and crocodylians, adding even small sums of cartilage impacts not only the length, but also the dorsoventral angulation and flexibility of the neck (Cobley, Rayfield & Barrett, 2013; Taylor & Wedel, 2013). The presence of larger intervertebral spaces in the posterior region of the neck indicates that this region could have less dorsoventral flexion due to the greater volume of synovial cartilage between the vertebrae (Cobley, Rayfield & Barrett, 2013; Molnar, Pierce & Hutchinson, 2014; Iijima & Kubo, 2019). The identification of variations in the thickness of the intervertebral synovial cartilage along the neck allows for more precise inferences of its range of motion, considering that the center of rotation is located in the intervertebral space (Haher et al., 1991; Selbie, Thomson & Richmond, 1993; Wintrich et al., 2019; Jones, Brocklehurstm & Pierce, 2021).

Considering the neck at rest, the partial overlap of the zygapophyses contributes to intervertebral flexibility, while the variation in the level of overlap along the neck indicate movement restriction in the caudal portion of the pterosaur neck (Cobley, Rayfield & Barrett, 2013; Iijima & Kubo, 2019). Bearing in mind that movement requires sliding between the zygapophyses, total cervical flexion in each direction could reflect an extremely reduced level of zygapophyses overlap between the most cranial vertebrae, as observed in extant archosaurs and mammals (Stevens & Parrish, 2005; Dzemski & Christian, 2007; Taylor, Wedel & Naish, 2009). In addition, the presence of synovial cartilage ensures greater mobility in the sliding movement of the zygapophyses, although a posture of full neck extension does not necessarily require a high sliding between the zygapophyses (Vidal, Graf & Berthoz, 1986; Vidal et al., 2020).

The relative verticalization of the of the neck at rest position of Azhdarcho lancicollis, relative to the more horizontal atlas and axis, is consistent with the neck posture of extant amniotes (Taylor, Wedel & Naish, 2009). The cervical posture observed in response to the downward tilt of the joint between the axis and the third vertebra and then the upward tilt presented by the joints of the mid-cervicals differs from the neck reconstruction in neutral posture for Azhdarcho lancicollis proposed by Averianov (2013). The reconstruction of the neck of Quetzalcoatlus by Padian et al. (2021) has a much more vertical neck in relation to the trunk in the “normal position” (as called by the authors), which was inferred based on the greater angles between the joints in the posterior portion of the mid-cervical series. However, we consider that the neck of Azhdarcho lancicollis would assume a position as vertical as suggested for Quetzalcoatlus by Padian et al. (2021) only during dorsiflexion. We believe that the reconstruction of the cervical series of Quetzalcoatlus may differ from our cervical reconstruction of Azhdarcho due to using different levels of overlapping of the zygapophyses and spacing between the vertebrae in this region, but the authors do not indicate how they defined the values used in their work (Padian et al., 2021).

The more vertical neck of Azhdarcho lancicollis in relation to the other pterosaurs analyzed probably received less compression and traction stress during the neck position at rest, as the head and neck are positioned closer to the center of mass (Dzemski & Christian, 2007; Marek et al., 2021). This lower mechanical demand may justify the reduction of the neural spines of mid-cervical vertebrae (Christian & Preuschoft, 1996; Tambussi et al., 2012). In contrast, the more horizontal necks in relation to the trunk observed in Anhanguera piscator and R. muensteri are consistent with the cervical vertebral architecture presented by both, which are adapted to resist higher static pressure (Dzemski & Christian, 2007; Taylor, Wedel & Naish, 2009).

Scars from soft tissue attachments are abundant in the analyzed pterosaur vertebrae, as expected in archosaur bones (McGowan, 1986). We interpreted many of them as muscle scars, although it is assumed that few muscles leave scars in extant archosaurs (Nicholls & Russell, 1985). The marked rough areas observed on the cranial and caudal surfaces of the neural spines of the analyzed birds and caiman were associated with the attachment of the ligamentum elasticum interlaminare in other extant archosaurs (Frey, 1988; Tsuihiji, 2004; Böhmer et al., 2020). The presence of a ligamentum elasticum interlaminare restricted only to the cranial and caudal ends of the neck in Azhdarcho lancicollis is also observed in birds with extremely long necks, as Rheiformes (Tsuihiji, 2004). Furthermore, the absence of the ligament in this region supports previous interpretations of less flexibility of the mid-cervical vertebrae of azhdarchid pterosaurs (Witton & Naish, 2008).

The presence of the ligamentum supraspinale has already been reported for extant crocodylians in the literature, without any mention of a ligamentum nuchae (Frey, 1988; Tsuihiji, 2004). Extant birds with bifid neural spines have the ligamentum elasticum interspinale, which is continuous and apparently has insertions similar to the ligamentum supraspinale of crocodylians (Tsuihiji, 2004; Tambussi et al., 2012). The medial grooves observed at the top of the neural spine of avian vertebrae have previously been related to the presence of the ligamentum nuchae, which would indicate an EPB level II inference for the presence of this ligament in pterosaurs (Wedel & Sanders, 1999). The continuous increase in the thickness of the top of the neural spines in the caudal portion of the neck may also be associated with the presence of developed muscles of the transversospinalis group (Snively & Russell, 2007). The variation we speculated in the height of the ligamentum nuchae between different pterosaurs has also been observed in neognaths that have it, although this ligament also varies in length in birds (Barkow, 1856; Baumel & Raikow, 1993; Yasuda, 2002).

Similarities in the attachment sites of the ligamentum supraspinale in alligators and the ligamentum nuchae in birds suggest that both may be homologous (Tsuihiji, 2004). However, ratite birds have both the ligamentum elasticum interspinale and ligamentum nuchae, both arranged dorsally to the cervical series and attached to the top of the neural spines (Tsuihiji, 2004). The inference of the presence of one or even two elastic ligaments placed dorsally to the cervical series is common for sauropodomorphs (Janensch, 1929; Alexander, 1985; Stevens & Parrish, 1999; Wedel & Sanders, 1999; Tsuihiji, 2004). The presence of the ligamentum supraspinale simultaneously with the ligamentum nuchae has already been inferred for an extinct crocodylian, but based on biomechanical requirements for cervical movement (Schwarz-Wings, 2014).

The above-mentioned ligaments could exert passive forces, stabilize the cervical series, and store energy (Dzemski & Christian, 2007). The robustness and elasticity of the inferred ligamentum elasticum interlaminare and ligamentum supraspinale could support additional weights to that of the head and help restore the neck to the rest position after ventral flexion (Gál, 1993; Tsuihiji, 2004; Witton & Naish, 2008; Tambussi et al., 2012; Naish & Witton, 2017). The presence of the ligamentum nuchae would probably contribute to the stabilization of the skull above the height of the trunk in a position at rest (Dimery, Alexander & Deyst, 1985; Tsuihiji, 2004). Particularly in Azhdarcho lancicollis, the inference of the ligamentum nuchae corresponds to the biomechanical requirement of a long neck with low vertebrae, which would represent a low bending moment and whose lack would make it unfeasible to support a long skull during ventral flexion (Christian & Preuschoft, 1996). The height of the ligamentum nuchae can be considered an extension of the height of the neural spine, contributing to the elongation of the lever arm of the bending moment produced along the entire neck during ventral flexion (Christian & Preuschoft, 1996). Furthermore, the ligamentum nuchae could be an additional surface for muscle attachments originating from the pectoral girdle (Dimery, Alexander & Deyst, 1985).

The robustness of the ligamentum collaterale and intervertebral thickness of cartilages in the posterior neck segment of pterosaurs probably contributes to the stability of this region, decreasing the range of motion and increasing stiffness between joints, thus increasing resistance to possible compression loads caused by dorsoventral flexion of the neck (Cobley, Rayfield & Barrett, 2013; Molnar, Pierce & Hutchinson, 2014; Iijima & Kubo, 2019; Moore, 2020). This resistance was probably further maximized in dsungaripteroids, including Anhanguera and Azhdarcho, by the formation of the notarium in osteologically mature individuals (Witton & Habib, 2010; Aires et al., 2021).

Implications for pterosaur foraging habits

The posture of the neck at rest inferred here is relevant to reconstruct head positioning during different foraging habits (Zweers, Bout & Heidweiller, 1994). The similarity of the anatomy and length of the skull of azhdarchids with birds that have terrestrial foraging habits indicates that they could be carried out by these pterosaurs, as proposed before (Witton & Naish, 2008; Witton & Naish, 2013; Bestwick et al., 2018). The more vertical neck in Azhdarcho lancicollis demonstrates that the head was higher than in previous reconstructions, which could contribute to the vision of the animal having greater scope in the environment (Dzemski & Christian, 2007). It has been argued that the presence of small muscles in a relatively long jaw in azhdarchids indicates that these pterosaurs possibly fed on small prey or carrion (Witton & Naish, 2008). The more vertical cervical posture in Azhdarchidae would minimize the stress received by the neck, which would allow these pterosaurs to lift small food items through their beak without causing high traction and compression on their vertebrae (Dzemski & Christian, 2007; Tambussi et al., 2012).

Anhanguerids are considered piscivores mainly due to their specialized rostral dentition and interaction with fish in the fossil record and isotopic analyses (Amiot et al., 2010; Tütken & Hone, 2010; Veldmeijer, Witton & Nieuwland, 2012; Wang et al., 2012; Bestwick et al., 2018). Anhanguerids would probably have been efficient at capturing fish at the surface of the water, a habit which would incur in low energy costs (Habib, 2015). Likewise, the Rhamphorhynchidae are mostly interpreted as piscivorous due to their conical and pointed teeth and the association with fish in fossils (Wellnhofer, 1975; Padian, 2008; Frey & Tischlinger, 2012; Hone, Habib & Lamanna, 2013; Bestwick et al., 2018). However, there are still inconsistencies regarding how fishing was carried out (Wellnhofer, 1975; Kellner & Tomida, 2000; Humphries et al., 2007; Frey & Tischlinger, 2012). The horizontal and slightly sinuous cervical posture inferred for Anhanguera piscator and Rhamphorhynchus muensteri is consistent with the piscivorous habits recognized for both, which should keep the head ventrally flexed to look down during foraging (Wellnhofer, 1975; Kellner & Tomida, 2000; Bestwick et al., 2018).

The cervical ligaments inferred here were likely important in stabilizing the spine and controlling the resting position of the pterosaur neck (Dimery, Alexander & Deyst, 1985; Gál, 1993; Dzemski & Christian, 2007). However, the inferred flexibility represents predicted cervical mobility, which may or may not represent movement control performed during the life of the pterosaur (Jones, Brocklehurstm & Pierce, 2021).

The excessive cervical stabilization guaranteed by the ligamentum collaterale next to the intervertebral joints and the ligamentum elasticum interlaminare between the neural spines corresponds to strategies for attacking prey in terrestrial or aquatic environments (Dimery, Alexander & Deyst, 1985; Gál, 1993; Ponseti, 1995; Tsuihiji, 2004). Furthermore, both ligaments could provide intrinsic stability to the neck in the flexed cervical posture or alert position (Dimery, Alexander & Deyst, 1985; Gál, 1993; Ponseti, 1995; Dzemski & Christian, 2007).

The ligamentum elasticum interlaminare and the ligamentum supraspinale could also help control movements by repositioning the neck to a rest posture after ventral flexion during foraging or hydration in pterosaurs (Dimery, Alexander & Deyst, 1985; Ponseti, 1995; Dzemski & Christian, 2007). However, such movements would not require energy expenditure and would not resist high loads, which indicates that these ligaments could lift the neck with little or no additional weight (Ponseti, 1995; Dzemski & Christian, 2007; Tambussi et al., 2012).

The ligamentum nuchae in quadruped animals functions by keeping the neck and head elevated (Dimery, Alexander & Deyst, 1985; Tsuihiji, 2004). Therefore, this ligament stabilizes the sinuous cervical posture above the trunk without energy expenditure. The height variation suggested for the ligament balances the conflict between the stress received by the neck and the establishment of cervical posture at varying angles during the range of movements (Christian & Preuschoft, 1996; Tambussi et al., 2012).

Conclusions

Pterosaur mid-cervical and posterior cervical vertebrae compose two distinct cervical anatomical segments, with longer mid-cervical vertebrae with more convex condyles and longer prezygapophyses, and shorter posterior cervicals with more developed vertebral processes. The presence of a fovea and extensive convex condyles, besides a rough area that borders all the articular surfaces of the vertebrae of the pterosaurs we analyzed, supports the presence of synovial cartilage in the vertebral joints, which are thicker near the base of the neck and probably affect the neck length and angulation at rest position.

The position of the neck at rest of the pterosaurs analyzed here indicates that they probably kept their heads above the level of the trunk and slightly flexed in a ventral direction. The verticalization of the neck in Azhdarcho lancicollis at rest is inferred by the orientation of the vertebrae above the main axis of their subsequent vertebrae, from the fourth to the ninth cervicals.

The osteological correlates observed in the vertebrae of pterosaurs support the presence of the ligamentum collaterale and ligamentum elasticum interlaminare as an EPB level I inference, the ligamentum elasticum supraspinale as an EPB level II inference and ligamentum nuchae are supported by an EPB level III inference. However, the recognition of the ligamentum nuchae of birds previously described in the literature indicates that the presence of this ligament in pterosaurs also represents an EPB inference level II. Probably, most ligaments were present throughout the entire neck, except for the ligamentum elasticum interlaminare, which may have been absent in the middle portion of the neck of Azhdarcho lancicollis, as suggested by the lack of osteological correlates. The scars and robustness of the neural spines indicate that the ligaments were probably more robust in the final segment of the neck. We discuss that the presence of these ligaments probably stabilized the cervical series, exerted passive forces, and stored energy.

Supplemental Information

Supplemental Information 1 Measurements referring to the cervical vertebrae of the pterosaurs analyzed

All measurements are in millimeters (mm)

We thank Niels Bonde and Maria Eduarda Castro Leal for their hospitality in Denmark, their help at the Zoological Museum of Denmark and for generously sharing the CT scans with us. We also thank the Head of the Geology section Michael Storey for allowing our analyses at the Zoological Museum of Denmark. We thank Alexander Averianov (Zoological Institute of the Russian Academy of Sciences, Russia) for generously sharing the laser scans with us. We thank Chisako Sakata (National Museum of Nature and Science, Japan) and curators Takanobu Tsuihiji (National Museum of Nature and Science, Japan) and Makoto Manabe (National Museum of Nature and Science, Japan) for the welcome and hospitality in Japan and for generously sharing the CT scans with us. We would like to thank Rodrigo Pêgas for the valuable discussions on the topic and for their help in photographing the specimens. We also thank the coordinators and employees of the Institute for Research and Rehabilitation of Marine Animals (IPRAM) (Cariacica, Brazil), the Caiman Project and the Marcos Daniel Institute (Vitória, Brazil) and the Veterinary Hospital of the Federal University of Espírito Santo (Alegre, Brazil). We thank Felipe Montefeltro (Universidade Estadual Paulista) and Fabiana Costa (Universidade Federal do ABC) for comments on an earlier version of this manuscript. We thank the editor Mathew Wedel and the reviewers Verónica Díez Díaz (Museum für Naturkunde), Michael Habib (University of California) and Felipe Pinheiro (Universidade Federal do Pampa) for their valuable review.

Anatomical Abbreviations

Ac ansa costotransversaria

arc articular capsule

bns base of the neural spine

cap carotid process

co cotyle

con condyle

ep epipophysis

faca facies articularis caudalis

facr facies articularis cranialis

fopn pneumatic foramen

fov fovea

ftr foramen transversarium

hyp hypapophysis

lcol ligamentum collaterale

liar ligamentum interarticulare

lilm ligamentum elasticum interlaminare

litc ligamentum intercostale

lnuc ligamentum nuchae

lsup ligamentum supraspinale

nc neural canal

ns neural spine

poex postexapophysis

poz postzygapophysis

prz prezygapophysis

ro roughness area

syn synovial cartilage

ta tuberculum ansae

tpr transverse process

tub tubercle

Additional Information and Declarations

Competing Interests

Author Contributions

Animal Ethics

Data Availability

The authors declare there are no competing interests.

Richard Buchmann conceived and designed the experiments, performed the experiments, analyzed the data, prepared figures and/or tables, authored or reviewed drafts of the article, and approved the final draft.

Taissa Rodrigues conceived and designed the experiments, authored or reviewed drafts of the article, and approved the final draft.

The following information was supplied relating to ethical approvals (i.e., approving body and any reference numbers):

The Institute for the Research and Rehabilitation of Marine Animals and the Marcos Daniel Institute provided full approval for this research under authorizations from SISFUNA (IEMA 001/2014, process 68077610; IEMA 001/2014, process 67277780) and SISBIO (34510 and 26896 for bird research and 92997549 for research with Caiman latirostris).

The following information was supplied regarding data availability:

The data is available at Morphosource. Our 3D models of the reconstructions of the cervical series in rest position, with ligaments, are available at:

Rhamphorhynchus muensteri, based on specimen MGUH 1891; https://doi.org/10.17602/M2/M588059 (Media ID 000588059);

Anhanguera piscator, based on specimen NSM-PV 19892; https://doi.org/10.17602/M2/M589271 (Media ID 000589271);

Azhdarcho lancicollis, based on several specimens; https://doi.org/10.17602/M2/M599131 (Media ID 000599131);

A 3D model of the main plate containing specimen MGUH 1891 (Rhamphorhynchus muensteri; Media ID 000588063) is available at https://doi.org/10.17602/M2/M588063 (Media ID 000588063).

3D models of the vertebrae of specimen NSM-PV 19892 (Anhanguera piscator) are available at:

https://doi.org/10.17602/M2/M589282 (Media ID 000589282);

https://doi.org/10.17602/M2/M589286 (Media ID 000589286);

https://doi.org/10.17602/M2/M589290 (Media ID 000589290);

https://doi.org/10.17602/M2/M589294 (Media ID 000589294);

https://doi.org/10.17602/M2/M589298 (Media ID 000589298);

3D models of the vertebrae of Azhdarcho lancicollis are available at:

https://doi.org/10.17602/M2/M568148 (Media ID 000568148);

https://doi.org/10.17602/M2/M568153 (Media ID 000568153);

https://doi.org/10.17602/M2/M568158 (Media ID 000568158);

https://doi.org/10.17602/M2/M568163 (Media ID 000568163);

https://doi.org/10.17602/M2/M568168 (Media ID 000568168);

https://doi.org/10.17602/M2/M568173 (Media ID 000568173);

https://doi.org/10.17602/M2/M568178 (Media ID 000568178);

https://doi.org/10.17602/M2/M568183 (Media ID 000568183);

https://doi.org/10.17602/M2/M568188 (Media ID 000568188).

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
