# Peer review of "Arthrological reconstructions of the pterosaur neck and their implications for the cervical position at rest"

_PeerJ, doi:10.7717/peerj.16884_

## Round 0.1 · original submission · Minor Revisions

All of the reviewers' comments seem justified and reasonable to implement. I look forward to seeing an improved version of this manuscript soon!

·

Basic reporting

The manuscript "Arthrological reconstruction of the pterosaur neck and their implications for the cervical position at rest" by the authors R. Buchmann and T. Rodrigues provides thorough osteological reconstructions and inferences on some soft tissues of several cervical remains from different pterosaurs. The text is clear, and the objectives are well reasoned, while the methodology is appropriate. Although I am not an expert on pterosaurs, I can see the authors handle a wide bibliography on these animals, as well as in the anatomy of extant archosaurs, and are able to use it profficiently through the text. The discussion and conclusions are also in concordance with the results.
I would only like to suggest increasing the quality of the figures, as some lines are difficult to discern (although it could also be the resolution in my computer). Besides, I think Figure 1 is missing. If that is the case, then the authors need to reorganise the rest and the citations in the manuscript. I also think that including a table with the measurements of all the cervical vertebrae (something the authors have already done, as indicated in the text) and another one with the EPB inferences between birds, crocodyles and pterosaurs could improve the final quality of the work.
I have also included some minor comments in the document, as well as a few more references that could be of interest to the authors.

Experimental design

The methods are sufficiently explained, like how the authors acquired the final 3D models or how they got access to the animals to dissect, how they did the dissections (e.g. what they were looking for), and how they have used this information in the osteological and soft tissues reconstructions of the necks of the pterosaurs.

Validity of the findings

As the authors state, these first reconstruction steps are of high importance for posterior biomechanical analyses, and they followed a well-reasoned red thread explaining how to proceed with these reconstructions so they are as accurate as possible.Also with the EPB information provided by the authors, this manuscript could become a reference in works on axial analyses of extinct organisms.

·

Basic reporting

The writing is generally strong - the text is descriptive, yet concise (possibly *too* brief in places, in fact - see below). The manuscript is generally well-referenced, as well. The only major concern I have with the basic reporting is that insufficient background and context is provided. The introduction leads off with how reconstruction of the neck in pterosaurs is difficult, which while true, does nothing to explain why it is worth doing. In a sense, this actually undervalues your hard work: pterosaur necks are *fascinating*, and you've done one of the most thorough looks at them to-date.

I recommend expansion of the earlier parts of the manuscript to introduce readers to the oddities of pterosaur necks. For a start, the cervical vertebrae in pterosaurs are nearly always (maybe even universally?) the largest elements in the spine. That is fantastically weird, not just within reptiles, but within vertebrates overall. I can't think of any similarly diverse or speciose clade where this is true. The necks are also supporting some of the most outrageous heads in the animal world. On top of all that, these crazy proportions of the head and neck are in a *flying animal*. The implications for center of mass shifting for flight control, among a half dozen other oddities of being a flighted creature with a massive neck, are worthy of consideration.

There is no need to be exhaustive and cover all of these features - I am merely providing some examples. But overall, something should be included to draw the reader into the question of "why are we interested in pterosaur necks?". If you set this up properly, you are likely to draw interest in the paper from folks beyond pterosaur specialists - and that is as it should be, because the work here is interesting beyond just pterosaur biology.

Experimental design

The research question is well defined, and it is clear how the work fills gaps in knowledge. It certainly seems as if the design is rigorous - the array of extant specimens used for comparison is robust. However, the Methods section needs some expanding to make it clear exactly how the phylogenetic bracket was applied. It is not sufficient for someone looking to reproduce your results to simply state that a bracket was used. I do honestly think you've done everything properly, it just needs a bit more detail in this regard so that other researchers can see exactly how you used your extant model species.

The methods for reconstructing unknown components of the skeletons are well described, and the source material is well explained. It is only the EPB that needs more explanation.

Validity of the findings

The conclusions are well stated and seem to be well supported, pending a bit more about the application of the phylogenetic bracket (see above). Overall, there is nothing that seems clearly "out of place" that isn't explained or any aspects of the reconstruction that do not seem to logically follow from the extant comparisons used. In short, the findings appear sound.

Additional comments

When this paper is fully revised and published, I hope the authors will consider advertising its presence not just among the science community, but also among the art community. The paleoart and concept art communities will find this study fascinating and useful.

·

Basic reporting

Buchmann and Rodrigues conducted a comprehensive assessment of the cervical arthrology of three pterosaurs from different species, which, when viewed from a phylogenetic perspective, are representative of reasonable diversity, including the early-diverging Rhamphorhynchus and two typically Cretaceous taxa. The work has several merits, with notable emphasis on the authors' careful dissection and first-hand illustration of the extant correlates they used to "calibrate" their EPB analyses. The paper is preceded by an introduction that competently contextualizes the issues addressed in the study. The MS is written in clear English and, it seems to me, without major grammatical problems. The figures are extremely relevant and aesthetically appealing (however, see comment below). The availability of raw data may be one of the very few issues that seem to affect the reproducibility of this well-designed and relevant study (also see below).

Experimental design

The authors conducted their study using a coherent experimental design and state-of-the-art methodologies. The research was directed towards addressing a clear knowledge gap, which was thoroughly explored throughout the text. The study is relevant and can provide insights into the lifestyle of an important group of vertebrates that lack perfect modern analogs. Methods are described in detail, but the reproducibility of the study may be compromised by the availability of raw data (see below).

Validity of the findings

The interpretations and discussions appear well-supported by the results. However, I miss a more in-depth discussion of the impact of the findings on the assumed lifestyles of the studied animals, especially considering that the chosen model groups have undergone paleobiological analyses leading to sometimes unconventional interpretations of their lifestyles. For instance, Rhamphorhynchus and Anhanguera have been interpreted as skimmers, a lifestyle intricately linked to cervical arthrology. Similarly, the interpretation of azhdarchoids as terrestrial stalkers could benefit from more direct corroboration within this line of research. I believe that a paleobiological connection of the results would not only enhance the study but also make it more relevant and citable.

Additional comments

As easily inferred from my comments above, the article appears relevant and well-written to me. I endorse its publication in its current form, with just a few minor suggestions that, I believe, would enhance the work:

- As I mentioned earlier, the paper would benefit from a clearer and more in-depth correlation of the results with the inferred lifestyles of the studied animals. Such a discussion would elevate the manuscript to another level, making it a reference not only for those interested in anatomy and refined biomechanics but also for anyone interested in the lifestyles of these most interesting animals.

- I'm not certain about PeerJ's policies regarding raw data availability. However, it appears to me that the reproducibility of the study depends on the availability of the three-dimensional models and reconstructions performed by the authors. I strongly suggest the authors consider making these materials available in an appropriate database.

- The images are particularly beautiful and well-crafted. Nevertheless, in order to enhance the contrast, I suggest using a black background for the three-dimensional models with light colors.

- I miss a table that displays all the inferred soft tissue structures, their presence/absence in extant correlates, and their degree of EPB reliability. This information, although scattered throughout the text, would be more easily accessed if clearly presented in a table.

---

## Round 0.2 · Minor Revisions

Thank you for your diligence in addressing the concerns of the reviewers. I'm happy with the current version, which I believe is ready for publication from a scientific perspective.

All that is left is to improve the reproducibility. It should be clearly stated where the CT raw data and model data can be accessed. Morphosource is mentioned but no additional data (e.g., permanent digital object identifier) is provided. It would be the customary to provided the raw image stack (if permission is allowed; if not possible it should be clearly stated how it can be obtained/assessed) and good scientific practice to provide the reconstructions in STL format. (See: https://royalsocietypublishing.org/doi/full/10.1098/rspb.2017.0194)

If you can include that information, there will be no remaining barriers to the acceptance of your manuscript for publication in PeerJ. Thank you in advance!

---

## Round 0.3 · Minor Revisions

Thank you for the information on data availability. That information information and the ark links should be included in the main manuscript, in the materials & methods. Once that information is included, I believe the manuscript will be acceptable for publication.

---

## Round 0.4 · accepted · Accept

Thank you for your patience and diligence with the revision process. This manuscript is ready for publication in PeerJ!